# Study investigating executive function in schizophrenia patients and their unaffected siblings

Feifei Xu[1], Zhenping Xian [2,3,4] *

**1** School of psychology, Zhejiang Normal University, Jinhua, China, **2** The First People's Hospital of Lianyungang, Lianyungang, China, **3** The Affiliated Lianyungang Hospital of Xuzhou Medical University, Lianyungang, China, **4** The First Affiliated Hospital of Kangda College of Nanjing Medical University, Lianyungang, China

* 2970782697@qq.com

**Data Availability Statement:** All relevant data are within the paper and its Supporting Information files.

**Funding:** The study was in part supported by Lianyungang First People's Hospital Youth Talent

## Abstract

### Introduction

Schizophrenia (SCZ) is characterized by widespread cognitive impairments, such as executive functions. Most of the available research indicate that executive impairment has a certain genetic predisposition. Shared neuropathological characteristics of patients with SCZ and their siblings may reveal intermediate behavioral phenotypes that can be used to further characterize the illness.

### Methods

Our study involved 32 SCZ patients, 32 unaffected siblings (US), and 33 persons as healthy controls (HCS). These three groups underwent a computerized version of the Wisconsin Card Sorting Test (WCST), and a battery of cognitive neuropsychological assessments. These tests also evaluate executive function and several cognitive domains.

### Results

The performed study on SCZ patients and their unaffected siblings showed an inferior WCST performance to the HCS subjects, further indicating that unaffected siblings have a functional impairment, and they also performed poorly on the neuropsychological assessment compared with the HCS.

### Conclusion

This result supports the claim that the development of functional impairment is not limited to SCZ patients and unaffected siblings may also have a certain level of abnormal brain function. Consequently. neurological abnormalities lead to the abnormal functioning in siblings and patients, suggesting that genetics plays a considerable role in such results.

Fund QN202109, The funders had no role in study design, data collection and analysis, decision to publish, or preparation of the manuscript.

**Competing interests:** The authors have declared that no competing interests exist.

# Introduction

SCZ is a serious genetic mental illness with both positive and negative symptoms [1,2], along with the disorganization. Cognitive impairment is a core feature of SCZ, which affects myriad cognitive domains [3]. Deficits appeared even before the onset of SCZ and patients' decline in later life is strongly related to their functional competencies [4]. An important issue in the study of SCZ is executive dysfunctions, that may affect other cognitive abilities [5,6]. Executive function, which is a higher-order and complex cognitive function, encompasses working memory, cognitive flexibility/set-shifting, and inhibition [7].

A large body of evidence suggest that the executive dysfunction is a core feature of SCZ resultant of altered neural mechanisms, which are related to its etiology and onset. It is also believed that the degree of executive dysfunction is attributed to the prognosis and functional responses of the individuals [8].

Brain regions such as dorsolateral prefrontal cortex (DLPFC) and ventrolateral prefrontal cortex (VLPFC) play an extremely important role in the development of executive function. Many brain imaging studies have shown that executive abnormalities are closely related to DLPFC regions [9]. Several studies have reported abnormalities in response inhibition in the anterior cingulate cortex (ACC) and PFC connections of unaffected siblings and SCZ patients [10].

Decades of family, twin, and adoption studies suggest a substantial genetic component to the risk of SCZ with some contribution from environmental influences. But transmission of SCZ is a complex process with a highly polygenic structure that is supported by the convergent evidence [11,12]. Moreover, both the patients and their unaffected siblings exhibit cognitive function impairments, such as social cognition, working memory, and attention [13,14].

The family members should be alerted when the effects of executive function impairment on individuals' cognitive and social functions are significant. Although the patients' siblings may not show symptoms, they are still vulnerable to the illness [15,16], and the possibility may be manifested at the early stages of executive impairment. If the disorder is further aggravated, it may lead to mental illness, and hence early assessment of siblings can help prevent psychiatric illness and detect it early [17].

In our study, we use a computerized version of the WCST to indirectly assess the participants' executive function. However, in many studies, participants have an older age at the onset [18]. To make the current findings more applicable, we initially selected patients with an earlier age of dysfunction onset along with the siblings who were in similar age. Several studies have suggested that patients with SCZ usually demonstrate similar age-dependent declines across most neuropsychologic functions [17,19]. Second, in accordance with previous studies, we focused on differences in WCST performance among patients, their unaffected siblings, and between these two groups. Second, consistent with previous research, we focus on performance differences on the WCST between patients and their unaffected siblings and between these groups [17,19]. We hypothesized that there might be a discrepancy between patients and their unaffected relatives regarding the executive impairment. Both the patients and their unaffected siblings showed an inferior performance compared with the HCS, while the unaffected siblings perform better than the patients. Our attention is more focused on younger participants since early screening and preventive measures for unaffected siblings are necessary.

# Materials and methods

## Participants

A total of 106 healthy control participants were recruited between September 2019 and May 2021, including 34 patients, 34 unaffected siblings of the patients, and 38 unrelated healthy control

subjects. The patients' symptoms were diagnosed by two psychiatrists of the Fourth People's Hospital of Hefei City [20]. In this study, the participants were selected according to the inclusion and exclusion criteria. Neuropsychological tests were conducted prior to the start of the study. Then, WCST is used to evaluate the execution function of participants. Also, all subjects received stable doses of atypical antipsychotic medication including risperidone, chlorpromazine, and olanzapine, while the remaining types of drugs would be converted to equal doses of risperidone. Noticeably, the Institutional Review Board of Anhui Medical University approved the study protocols. Accordingly, all participants provided written informed consent. All participants were blinded to the purpose of the study and were in good health before the test.

Moreover, the patients were recruited from the clinic of the Fourth People's Hospital of Hefei City, and the test subjects were selected according to the enrollment and exclusion criteria. The enrollment criteria were as follows: (1) meeting the diagnostic criteria of the DSM-IV while the patients' symptoms were evaluated by two psychiatrists; (2) there was no frequent adjustment of drugs and doses for at least two months before participating in the study; (3) no obvious visual impairment [21]; (4) The patient does not have cognitive disorders [22] (5) They have not participated in any other clinical trials. The exclusion criteria were as follows: (1) severe head trauma or history of neurological diseases, serious physical illness, uncooperative persons, etc. [23]; (2), the patient does not have substance abuse within the past six months [24]; (3) unstable emotional state or having a Hamilton Anxiety/Depression Rating Scale (HAMA/HAMD) score > 7 points. All patients were treated as clinically stable for 2 weeks without drug replacement or rehospitalization (Fig 1).

The unaffected siblings were brothers or sisters of the patients. Furthermore, the inclusion criteria for the siblings were as follows: (1) no mental disorder according to the DSM-IV; (2) intact general cognitive function according to the Beijing Version of the Montreal Cognitive Assessment (MoCA) test scores > 22 points; and (3) HAMA or HAMD < 7 points. The exclusion criteria were as follows: (1) a lifetime history of substance abuse; (2) history of head trauma, neurological diseases, or other serious physical illnesses; and (3) a history of the long-term use of drugs or mental disorders. The criteria for selecting healthy subjects were the same as those used for siblings. The healthy subjects were recruited from nearby communities for comparison purposes and accorded with the patients considering gender and age. There are no cognitive impairments or mental problems among the healthy subjects. All participants had the visual acuity at normal levels.

## Neuropsychological assessment

Standardized neurological tests were used to investigate the participants' basic cognitive status and emotions, such as anxiety and depressive symptoms. Then, a battery of neuropsychology assessments were performed during the subjects' first visit in order to evaluate their global cognition, emotion, memory, and attention [25]. These assessments consisted of the Montreal Cognitive Assessment (MoCA, Beijing-version), cronbach's α = 0.836, Hamilton Anxiety Rating Scale (HAMA), cronbach's α = 0,93, Hamilton Depression Rating Scale (HAMD), cronbach's α = 0.92, digital span test (forward/backward) (DST F/B), Stroop Test (color/word/ interference)(SC/W/I/ T), and Trail Making A/B (TMTA/B). Moreover, the patients' symptoms were assessed using the Positive and Negative Symptoms Scale (PANSS), cronbach's α = 0.779. These tests were evaluated by experienced psychologists and psychiatrists. After the tests were completed, all participants were briefed on the results.

## Wisconsin Card Sorting Test (WCST)

Executive function is measured by the WCST, which has been employed to assess the patients and the HCS. The computerized version of WCST is used to measure executive function,

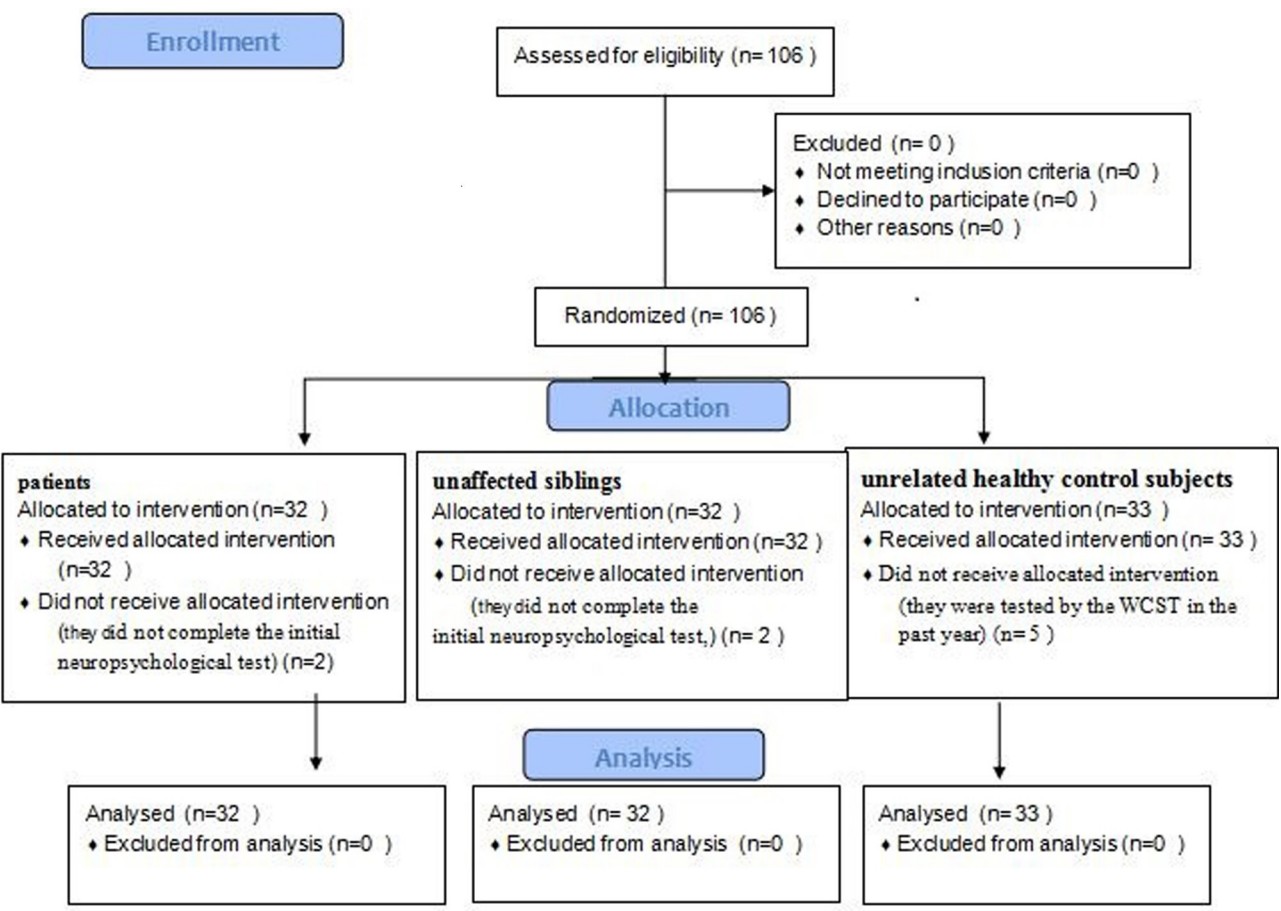

**Fig 1. Flow chart.**

mainly the ability to abstract reasoning and transform cognition and concept formation. The WCST displays 4 stimulus cards, and a response card. This study counts the total number of correct responses (TC), the total number of wrong responses (TE), the number of persistent responses (PR), the number of persistent errors (PE), and the number of trials to complete the first category (TCFC) [26,27]. Therefore, the larger the number of TC, the stronger the abstract generalization, working memory, attention, and executive control ability [27]. Also, larger TE and PE numbers, which reflect the cognitive transfer and executive control functions of the subjects, indicate the reduction of cognitive flexibility. Noticeably, the larger number of completed classifications, lead to a greater concept conversion, classification initiative, and diversity concept [28].

## Statistics

A data analysis was performed using IBM SPSS, including the one-way analysis of variance (ANOVA) in order to compare the demographics across the three groups of participants. Furthermore, $\chi^2$ analyses were used to compare the gender distribution among the participants. The results of the WCST were analyzed as a between-subjects variable by ANOVA corresponding to all groups. In addition, age and gender were used as the covariates. Afterwards, pairwise comparisons were performed using Fisher's least significant difference to quantify the presence of significant differences or interactions among participants. Notably, the degree of

correlation between variables was analyzed by Pearson's correlation. The test scores are expressed as an averaged value with their corresponding standard errors, and standardized effects were calculated using Hedge'g. Thus, the significance level was set at a 2-tailed $P$ ($\alpha$ = 0.05/3) to account for type I errors.

## Results

The current data comprises the analyses results of 97 participants. Because, two patients and two unaffected siblings did not complete the initial neuropsychological tests, and hence the WCST was not completed. The demographics and corresponding clinical information of the participants are listed in Table 1. Similarly, the clinical symptoms and neuropsychological assessments are reported in Table 1. Taking into account their gender and education level, schizophrenic patients, their unaffected siblings, and HCS required approximately 2 hours to complete all tests, such as neuropsychological tests and WCST tasks.

### Demographic characteristics, clinical information, and neuropsychological tests of three groups

As reported in Table 1, the average age of the 32 patients (17 females and 15 males) with SCZ is 22.4 years, and on average, the patients had 14.46 years of education. Thirty-two (15 females

**Table 1. Demographic characteristics, clinical information, and neuropsychological tests of three participating groups.**

| Characteristics | SCZ patients ($n$ = 32) | Unaffected siblings ($n$ = 32) | HCS ($n$ = 33) | $F/\chi^2$ | $p$ | $\eta_p^2$ |
|---|---|---|---|---|---|---|
| Gender (M/F) | 15/17 | 17/15 | 17/16 | 0.270 | 0.874 | |
| Age (years) | 22.40±4.28 | 23.03±2.19 | 22.24±2.53 | 0.569 | 0.568 | 0.012 |
| Education (years) | 14.46±2.04 | 14.21±2.21 | 14.03±2.18 | 0.342 | 0.712 | 0.007 |
| Total PANSS | 45.28±9.47 | NA | NA | NA | NA | NA |
| PANSS Positive | 10.43±2.81 | NA | NA | NA | NA | NA |
| PANSS Negative | 10.44±2.61 | NA | NA | NA | NA | NA |
| Illness duration (years) | 5.46±4.29 | NA | NA | NA | NA | NA |
| Risperidone equivalent (mg) | 5.71±1.68 | NA | NA | NA | NA | NA |
| HAMA | 4.09±.090 | 4.84±.846 | 4.33±.333 | 0.837 | 0.436 | 0.018 |
| HAMD | 3.75±.751 | 4.65±.650 | 4.24±.248 | 1.550 | 0.218 | 0.032 |
| MoCA | 25.31±3.14[b] | 25.93±2.47[c] | 29.51±1[b,c] | 29.817 | <0.001 | 0.391 |
| VFT | 18.65±4.08 | 18.93±4.08 | 20.33±4.08 | 1.218 | 0.3 | 0.026 |
| DSTF | 7.436±1.01[b] | 7.5±0.83[c] | 8.36±0.96[b,c] | 9.852 | <0.001 | 0.175 |
| DSTB | 5.21±1.21[b] | 5.46±1.04[c] | 6.42±1.60[b,c] | 7.463 | 0.001 | 0.138 |
| SCT | 18.51±5.71·[b] | 15.97±5.03 | 13.41±3.39[b] | 9.119 | <0.001 | 0.164 |
| SWT | 22.26±8.55[b] | 18.77±5.77[c] | 14.71±4.55[b,c] | 10.983 | <0.001 | 0.191 |
| SIT | 33.21±12.06[b] | 29.04±9.26[c] | 21.19±5.91[b,c] | 13.706 | <0.001 | 0.288 |
| TMAT | 52.77±18.68[b] | 46.15±17.01[c] | 35.58±10.86[b,c] | 9.612 | <0.001 | 0.171 |
| TMTB | 112.89±41.63[b] | 95.4±30.22[c] | 71.49±17.43[b,c] | 14.343 | <0.001 | 0.238 |

Note: HCS, healthy controls; M, male; F, female; PANSS, Positive and Negative Syndrome Scale; NA, not applicable; HAMA, Hamilton Anxiety Rating Scale; HAMD, Hamilton Depression Rating Scale; MoCA, Montreal Cognitive Assessment Test; VFT, Verbal Fluency Test. SCT: Stroop Color test; SWT: Stroop Word test; SIT: Stroop Interference test; DSTF: Digit span (forward); DSTB: Digit span (backward); TMTA: Trail Making A; TMTB: Trail Making B.

a, SCZ vs unaffected siblings significantly different (p <0.017).

b, SCZ vs HCS significantly different (p <0.017).

c, unaffected siblings' vs HCS significantly different (p <0.017).

and 17 males) unaffected siblings ($M_{age}$ = 23.03, $M_{education}$ = 14.21) were recruited. The siblings are all the patients' brothers or sisters. The HCS group included 33 participants (16 females and 17 males) with an average age of 22.5 years and 14 years of education on average. No significant differences was found among these groups in terms of age, education, and gender.

However, there was a significant difference among the patients, sibling, and HCS in certain parts of the neuropsychological tests, such as SCT, SWT, SIT; DSTF, DSTB; and TMT A and B. Strikingly, post hoc analyses revealed significant differences between the patients and HCS scores as follows: SCT [Hedge's g = 1.08,p<0.001, 95% CI(2.75,7.52)], SWT [g = 1.09, p<0.001,95% CI = (4.33,10.79)], SIT [g = 1.25, p<0.001, 95% CI = (7.35,16.71)]; DST F[g = -0.92, p<0.001, 95% CI = (-1.36,-0.46)], DSTB [g = -0.82,p<0.001, 95% CI = (-1.81,-0.54)]; and TMT A [g = 1.11,p<0.001, 95% CI = (9.15, 24.68)], and B [g = 1.28, p<0.001, 95% CI = (25.89,56.31)]. In these tests, the patients' performance was also inferior to the control groups. Compared with patients, there was no significant differences in the siblings considering these variables. SWT [g = 0.77, p = 0.013,95% CI = (0.90,7.39)], SIT [g = 1, p = 0.001,95% CI = (3.26,12.66)], DSTF [g = -0.94, p<0.001,95% CI = (-1.33,-0.42)], DSTB [g = -0.70, p = 0.005,95% CI = (-1.55,-0.28)], TMT A [g = 0.73, p = 0.0011,95% CI = (2.45,18.06)] and TMT B [g = 0.96, p = 0.001,95% CI = (9.87,40.21)], indicating differences between the HCS and unaffected siblings, while there was no significant difference in SCT between these groups [g = 0.59,p = 0.036, 95% CI = (-0.37,5.51)]. Some differences were also found in the other variables such as the MOCA total score, where in the patients with SCZ had lower scores than the HCS. However, the HAMA and HAMD scores are lower than those of the HCS. Furthermore, all participants' test scores are within the scope of the study requirements. In addition, the HAMA, HAMD, and VFT scores did not significantly differ among the participants.

## Assessment of executive function

The data displayed in Fig 2 show the results of the WCST. Accordingly, executive function performance, the total correct responses, the total errors, persistent responses, and persistent errors significantly differed among the three groups as follows: TC[$F_{(2,94)}$ = 4.421,p = 0.015, $\eta_p^2$ = 0.087], TE[$F_{(2,94)}$ = 17.267,p<0.001,$\eta_p^2$ = 0.271], PR [$F_{(2,94)}$ = 4.571,p = 0.013,$\eta_p^2$ = 0.09], PE [$F_{(2,94)}$ = 5.945,p = 0.004,$\eta_p^2$ = 0.113]. Moreover, the post hoc analyses show differences in four variables between the patients with SCZ and the HCS as follows: TC [g = -0.60, p = 0.009,95% CI = (-16.49,-2.41)],TE [g = 1.41, p<0.001,95% CI = (11.75,23.86)], PR [g = 0.61,p = 0.008,95% CI = (2.15,13.74)] and PE [g = 0.89,p = 0.002, 95% CI = (3.34,17.27)]. Furthermore, there are significant differences between the unaffected siblings and the HCS as follows: TE [g = 0.88,p = 0.004,95% CI = (2.86,15.03)], PR [g = 0.91,p = 0.016, 95% CI = (1.36,13.02)] and PE [g = 0.67,p = 0.01, 95% CI = (1.8,12.79)], while there was no significant differences in TC between these groups (g = -0.51,p = 0.017, 95% CI = (-17.31,0.057)]. By comparing the siblings with the patients, we found that TE [g = 0.63,p = 0.005, 95% CI = (-14.99,-2.73)] has significant differences. In addition, the trial to complete the first category did not reveal significant differences (F = 1.175, p = 0.313, $\eta_p^2$ = 0.025) among these groups as shown in Fig 2.

## Correlation analysis between WCST and neuropsychological tests in patients and unaffected siblings

The results also show that there is a significant correlation between the trial to complete the first category, and all neuropsychology indicators. The trial to complete the first category has a strong negative correlation with MOCA (r = -0.27, p = 0.031) and TMT B (r = -0.27, p<0.001).

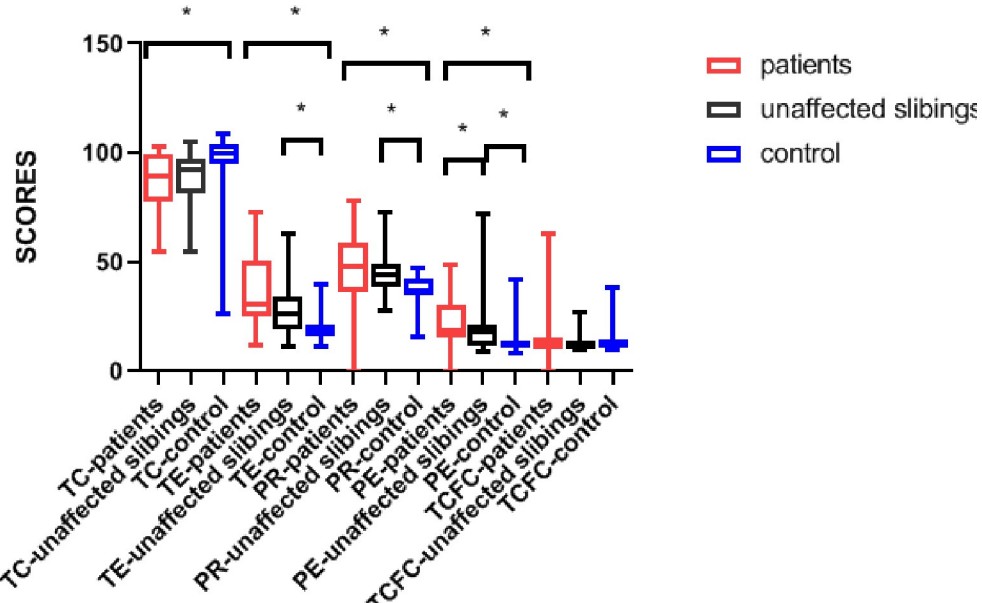

**Fig 2. Assessent of executive functions for SCZ, Unaffected siblings and HCS groups.** TC = the total number of correct responses; TE = the total number of error responses,PR = the number of persistent responses;PE = the number of persistent errors; TCFC = the number of trials to complete the first category;* Significant at P<0.017 ((α = 0.05/3)).

Furthermore, there is a strong positive correlation with SCT (r = 0.424, p<0.001), SWT (r = 0.297, p = 0.017), and SIT (r = 0.402, p<0.001). Conversely, no significant correlation was found between the remaining aspects of the WCST (TC, TE, PE and PR), and the neuropsychological tests as can be seen in Table 2.

## Correlation analysis between WCST and Clinical symptoms in the patients with SCZ

These correlation results show that the trial to complete the first category has a strong negative correlation with PANSS total (r = -0.403, p<0.01), PANSS positive (r = -0.351, p<0.01), PANSS negative (r = -0.290, p<0.05), and illness(r = -0.342, p<0.05). The total correct responses are strongly negatively correlated with PANSS total (r = -0.401, p<0.01), PANSS positive (r = -0.302, p<0.05), and the age. Noticeably, the age is significantly correlated with total correct responses (r = -0.310, p<0.05), persistent responses (r = -0.317, p<0.05), and the trial to complete first category (r = 0.324, p<0.05) as reported in Table 3.

**Table 2. Correlation analysis between WCST and neuropsychological tests in patients and unaffected siblings.**

| Characteristics | total correct | total errors | persistent responses | persistent errors | Trial to complete first category |
|---|---|---|---|---|---|
| MoCA | 0.107 | -0.024 | -0.005 | 0.036 | -0.27[a] |
| Stroop Color test | -0.066 | 0.148 | 0.024 | 0.090 | 0.424[b] |
| Stroop Word test | -0.031 | 0.079 | -0.086 | -0.007 | 0.297[a] |
| Stroop Interference test | -0.070 | 0.071 | -0.157 | -0.123 | 0.402[b] |
| Trail Making A | 0.097 | 0.040 | -0.005 | -0.029 | -0.025 |
| Trail Making B | -0.141 | 0.247 | 0.034 | 0.070 | 0.124 |

Note:

a: p<0.05

b: p<0.01.

**Table 3. Correlation analysis between WCST and Clinical symptoms in patients with SCZ.**

| Characteristics | total correct | total errors | persistent responses | persistent errors | Trial to complete first category |
|---|---|---|---|---|---|
| PANSS positive | -0.302[a] | 0.069 | -0.060 | 0.215 | -0.351[b] |
| PANSS negative | -0.114 | 0.107 | -0.143 | 0.104 | -0.290[a] |
| PANSS total | -0.401[b] | 0.210 | -0.025 | 0.382[a] | -0.403[b] |
| age | -0.310[a] | 0.081 | -0.317[a] | 0.186 | 0.324[a] |
| education | 0.113 | -0.362[a] | 0.113 | -0.107 | 0.018 |
| Illness | 0.080 | 0.023 | -0.084 | 0.136 | -0.342[a] |

Note:

a: p<0.05

b: p<0.01.

## Discussion

In the current study, SCZ patients and their unaffected siblings showed inferior WCST performance than the HCS, further indicating that unaffected siblings could have functional impairment, since they also performed worse on the neuropsychological assessment than the HCS. The test results of the patients and their siblings show that there is a strong negative correlation between the TCFC and MOCA scores. along with a strong positive correlation between the TCFC and Stroop tests, suggesting that not only the patients, but also their unaffected siblings may have impairments in executive function i.e., PR and PE.

Patients with SCZ often tend to have executive dysfunction. It is to be emphasized that executive function is an advanced cognitive function, while cognitive dysfunction is a manifestation of a series of SCZ independent positive and negative symptoms, which affect patients' daily life, social interaction, employment, etc. [29]. Related studies have also reported that cognitive dysfunction often precedes other symptoms in patients [30], suggesting that cognitive deficits may be a hallmark of neurodevelopmental abnormalities, and are somewhat associated with heredity [31]. Therefore, patients' unaffected siblings may also develop functional impairment.

Studies have pointed out that executive function gradually declines with age [32,33]. The heritability of executive dysfunction has also been confirmed in previous studies [15]. But the average age of participants in many of those studies is around 40 years old [34,35]. Our results confirm that at the averaged early ages, patients and their siblings have executive function impairment, This result leads us to believe that being a relative of a SCZ patient may be a risk factor for the disruption in cognitive abilities or low performance. When intervening with family members of SCZ patients (psychoeducation, social skills training, support to reduce expressed emotion, case management, etc., it must be taken into consideration that some primary caregivers may also suffer from some degree of executive dysfunction [36].

Our research confirms that there is a negative correlation between executive performance and the age in the patients' group. Without excluding age as a factor, we were unable to determine whether functional impairment in patients and siblings was affected by age [37]. Therefore. we selected a younger group in order to avoid age interference factor. The results are also consistent with previous studies, confirming that at early ages, patients and their unaffected siblings may have executive function impairment. The patients and their unaffected siblings showed worse WCST performance than the HCS, further indicating that unaffected siblings could have functional impairment, since they also performed worse on the neuropsychological assessment than the HCS [38,39].

Cognitive impairment is widely recognized as a core feature of SCZ and is closely related to long-term functional outcomes [40]. Executive function refers to the ability to establish goals, make and revise plans, and carry out purposeful activities. Demonstrating functional impairment is a key dysfunction in patients with SCZ [41]. In our study, the neuropsychological assessment data suggest that patients with SCZ have relatively lower scores than the HCS, suggesting that patients with SCZ have impairments in their cognitive function, attention, and cognitive flexibility. Other studies have also found cognitive impairment in SCZ [42]. As expected, the patients' executive function is also significantly inferior to the studied HCS.

Evidence from several studies suggests that reduced activating abnormalities in the PFC are associated with the development of executive functional deficits in patients with SCZ [43,44]. A large number of converging evidence indicate that abnormalities in the DLPFC are the initial inducers of higher-order cognitive processing in the pathophysiology of SCZ [45]. Notably, the results of the WCST indicate that the gray matter volume in the DLPFC was significantly reduced in cognitively impaired patients [46]. Furthermore, this result was consistently observed in related imaging studies involving patients with SCZ [47]. Therefore, the DLPFC plays a very important role in the WCST results and could be related to persistent responses and persistent errors [48].

Our study also suggests that schizophrenic patients have a poor performance in the WCST. The patients have lower TC scores, suggesting that patients may have a certain cognitive dysfunction as cognitive dysfunction affects the overall score, processing speed, attention/vigilance, short-term memory, etc [49]. This finding is also consistent with our results as the patients' MOCA scores are significantly lower than those of the HCS [50]. The total error scores of the patients with SCZ is significantly higher than the HCS, showing that there is a poor cognitive flexibility [51]. Cognitive flexibility bias also implies that patients have a poor ability to employ strategies, which, in turn, affects their daily activities, such as multitasking and finding new, adaptable solutions for problems [52].

Patients with SCZ usually have abnormalities in persistent responses, which is one of the best indicators of all WCST brain injury indicators, and whether focal damage exists in the frontal lobe or not. This finding also suggests that the abnormal brain function is the cause of the poor performance [53]. Larger numbers of persistent errors made by the patient show problems in concept formation, such as the use of appropriate corrections and concept adherence due to brain frontal lobe impairment [54]. In recent years, there have been reports of poor patient performance in the WCST. Similarly, the studies patients also showed significantly worse WCST performance than the HCS, which is consistent with previous results indicating that the impaired cognitive ability involves with the performing function in SCZ [55,56]. We also found that the patients' trial to complete the first category pose no abnormalities, indicating that the patients' abstract generalization ability was not impaired. There are occasions indicating that the patient's abstract generalization ability was recovered to some extent during the recovery period. Compared with the previous studies, our HCS performed normally, and patients did not differ much from the HCS. Usually, patients should act abnormal on this indicator, but this current observation does not match well with the previous results [57]. We found that this observation could be related to the patients' course of illness and education, since a higher educational level also has a great impact on this result. Accordingly, cognitive dysfunction is not particularly prominent in patients with higher functional levels and education [58]. Furthermore, the stability of the outpatient's medications, the duration of the disease, and the dose of antipsychotics may differ, which may cause the discrepancy from previous results. We chose stable outpatients who had not changed their medications within eight weeks. in order to ensure the reliability of the results both the patients and the HCS were recruited with comparable educational level.

According to the current findings when participants' average age is lower, patients and their unaffected siblings performed poorly on the WCST, suggesting that executive dysfunction is not only limited to patients but rather greatly related to the heredity and family [59]. Patients have impairments in certain abilities, such as cognitive flexibility and abstract generalization skills; thus, when completing WCST tasks, there will be more persistent errors, leading to difficulty in achieving high scores. By exploring these similarities while comparing patients with their unaffected siblings, primary evidence of a specific phenotype is emerged. The unaffected siblings performed slightly better that the patients on the WCST but their performance significantly differed from the HCS [60]. This finding also indicates that unaffected siblings could have impairments in certain functions, such as reaction inhibition, short-term memory, attention, and abstract generalization. In a family, a certain type of functional damage is related to the heredity to a certain extent and eventually becomes a factor affecting the illness intensity [61]. An abnormality in executive function usually precedes patients' psychiatric symptoms [62], and cognitive impairment accelerates the development of symptoms and eventually leads to illness. These defects cause social or occupational dysfunctions, and hence a low-quality life. In addition, in our study, both the Stroop test and MOCA were significantly associated with the WCST. This relationship also indirectly demonstrate that the employed WCST task is consistent with the classic Stroop test, and both can evaluate the execution function. This result further affirms the existence of cognitive dysfunction between the patients and their unaffected siblings.

Some limitations of this study should be noted. First, we used the WCST and a series of neuropsychological tests to investigate the existence of executive control defects. This analytical strategy enabled us to obtain supporting evidence of our hypotheses. However, the results might show some inaccuracy due to some subjective factors. Therefore, behavioral and physiological studies should be performed to provide more reliable findings. Second, compared with previous studies using the WCST, our research employed relatively fewer indicators; thus, the results need to be further expanded [43,63]. Third, the average age of the unaffected siblings in our sample is 23 years, suggesting that some siblings may have not passed the risk of SCZ, which may also result in some disagreements with previous studies. Fourth, the drugs and dosages used by the patients have not been described in detail, which is also a deficiency of this study. In the future, further detail about the prescribed medications can be provided. Fifth, our study size is relatively small and may affect the final result. Finally, several patients were still in the treatment phase, which may affect our results as some antipsychotics may be associated with WCST performance. In future research, we plan to combine neuroimaging to explore variations in the participants' brains. Furthermore, the age distribution of participants will be expanded to maximize the scalability of findings.

## Conclusions

In summary, at a younger age, the patients with SCZ and their unaffected siblings perform poorly in the WCST confirmed by the performed neuropsychological tests. The patients and their siblings have also significant correlations with general cognitive tasks in WCST scores. We found that both the patients and their siblings have cognitive function damage compared with the HCS, while the unaffected siblings performed better than the patients. These findings also indicate that, at the younger ages, siblings may already have executive impairment. Unaffected siblings could also have a certain level of abnormal brain function. Finally, neurological abnormalities lead to abnormal functioning in patients and their siblings, suggesting that genetics plays a considerable role in achieving such results.

## Supporting information

**S1 Data.**
(XLS)

## Author Contributions

**Data curation:** Feifei Xu.

**Formal analysis:** Feifei Xu.

**Investigation:** Feifei Xu.

**Methodology:** Feifei Xu.

**Writing – original draft:** Feifei Xu.

**Writing – review & editing:** Zhenping Xian.

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
