## [Decision Letter · Decision Letter 0]

1 Dec 2022

PONE-D-22-29812Study investigating executive function in schizophrenia patients and their unaffected siblingsPLOS ONE

Dear Dr. xian,

Thank you for submitting your manuscript to PLOS ONE. After careful consideration, we feel that it has merit but does not fully meet PLOS ONE’s publication criteria as it currently stands. Therefore, we invite you to submit a revised version of the manuscript that addresses the points raised during the review process.

Thank you for submitting your valuable work.

The reviews, which are insightful and interesting, pointed to some unexplained aspects. The authors will notice the reviewers found merits in your study, but also raised several important concerns.

By my own reading, the manuscript still needs a lot of refinement, mostly related to soundness, conciseness and the control of confounding factors. Although this may sound counterintuitive, I am keen to understand authors' claims and keen on reading a refined manuscript. Please note the reviewers' criticism concerning writing style, English, stats and uncontrolled aspects of your study.

Please respond to the comments AND highlight all changes.

We look forward to receiving your revised manuscript.

Kind regards,

Thiago Fernandes, EbS, MA, Sp. Neuro PhD

Academic Editor

PLOS ONE

Journal Requirements:

2. PLOS ONE does not copy edit accepted manuscripts (https://journals.plos.org/plosone/s/criteria-for-publication#loc-5). To that effect, please ensure that your submission is free of typos and grammatical errors.

**Additional Editor Comments:**

Please read my comments.

Reviewers' comments:

Reviewer's Responses to Questions

**Comments to the Author**

1. Is the manuscript technically sound, and do the data support the conclusions?

Reviewer #1: Yes

Reviewer #2: Yes

Reviewer #3: Partly

2. Has the statistical analysis been performed appropriately and rigorously? 

Reviewer #1: No

Reviewer #2: No

Reviewer #3: Yes

3. Have the authors made all data underlying the findings in their manuscript fully available?

Reviewer #1: Yes

Reviewer #2: Yes

Reviewer #3: Yes

4. Is the manuscript presented in an intelligible fashion and written in standard English?

Reviewer #1: No

Reviewer #2: No

Reviewer #3: No

5. Review Comments to the Author

Reviewer #1: This is a very interesting and timely study. I think it was well-conducted and well-written. Although I have no several concerns, a few aspects need to be addressed. Hope the authors find the suggestions useful.

1 ) Please enlist a native English speaker or a colleague to edit the manuscript;

2 ) Although this is a comment for the Abstract, this should be considered thorough the entire text: the authors need to improve readability and conciseness. The Abstract is way too long and both methods and results subsections need to be shortened. This will improve the reading but will also leave the impression of a "punchy" abstract. Please do consider this throughout the text and try to shorten some excessive details (e. g. the first paragraph of Introduction);

3 ) In this line, the authors also need to give more details on the fundamentals of their rationale. The Introduction needs to cover (1) definitions, (2) past findings, (3) the main gaps, (4) how the authors are willing to cover these gaps, (5) the main hypotheses and objectives, and (6) a "punchy" statement at the end of the Introduction pointing to the study;

4 ) I'd highly suggest the authors to consider a few references for debating the connectivity (e.g., 10.1093/schbul/sbm034 or 10.1016/j.nicl.2014.07.003) and the aspects of physiology (10.1016/j.psychres.2021.114200);

5 ) Regarding the Methods, more details are need. For example, the authors need to explain the timeline, screening, confounding variables, the Cronbach values for each scale, all details of the stats. More specifically, details on eligibility is needed: were the patients free of cognitive disorders? (10.3389/fpsyt.2013.00182), had normal visual acuity (10.1016/j.jpsychires.2022.03.014 and 10.1016/j.clinph.2005.06.013), had no previous contact with substances (10.3389/fpsyg.2018.00288), were drug free or what kind of specific meds were taking (10.1016/j.schres.2018.09.002 / 10.31887/DCNS.2006.8.1/bwinklbaur / 10.1016/j.schres.2017.04.016), if BDNF or GDNF levels were collected and why, if they had previous contact with any clinical trial (10.1097/00004714-200012000-00019) and the equivalents for the meds, if any;

6 ) Use more illustrative graphs to demonstrate data;

7 ) Use effect sizes and CIs for the analysis;

8 ) Use the corrected p-value;

9 ) Explain that data are presented as Mean + SD, for example;

10 ) Please bear that boxplots with individuals values are more representative of the data and the sample (10.1038/nmeth.2813)

11 ) Details of meds should be place in Methods;

12 ) Please bear that the use of effect sizes and CIs are needed for the frequentist analyses;

13 ) My main concern regarding the Discussion is the lack of tightening it up to the findings, but most importantly, the lack of references. There are several sentences without a reference. This definitely needs to be rewritten or adjusted;

14 ) Please update the references list using DOI accordingly to the Journal's standards;

Reviewer #2: Xu et al. investigated differences in executive functions assessed by WCST among 32 patients with schizophrenia, 32 unaffected siblings, and 33 healthy controls. I guess the authors want to claim that ‘Among executive functions, total error (TE) was different among three groups. The TE was not correlated with other cognitive neuropsychological assessments or clinical variables.’ However, there were many careless mistakes, such as typos, inappropriate spaces, inappropriate periods and casual or inappropriate terms (e.g., sick, injury, damage, so, good health), throughout manuscript. After correcting these mistakes carefully, the authors should resubmit the manuscript. Furthermore, I have some suggestions that may, in my opinion, further improve the quality of the paper:

There were some concerns as follows;

1. There were many careless mistakes, such as typos, inappropriate spaces, inappropriate periods and casual terms, throughout manuscript.

2. The authors should focus on executive functions but not other cognitive functions or cognitive neuropsychological assessments throughout manuscript.

3. The authors should focus on TE among executive functions throughout manuscript as TE was only different among three groups.

4. In Abstract ‘the WCST results demonstrate that the total correct (TC), total error (TE), perseverative response (PR) and perseverative error (PE) scores in the SZ group were significantly lower than those in the HC group (TC (p=0.011), TE (p<0.001), PR (p=0.007) and PE (p=0.002)), and compared to the unaffected siblings’, These directions were incorrect, e.g., TE in the SZ group was significantly lower than those in the HC group?? Definitely incorrect.

5. In Abstract ‘Neurological abnormalities lead to abnormal functioning in siblings and patients, suggesting that genetics plays a considerable role in such results.’ The current study did not examine or focus other cognitive function except for executive function.

6. In introduction ‘Most research indicates that executive impairment is a core feature of cognitive impairment [4].’, Please clarify why the author consider executive impairment is a core feature among cognitive impairments.

7. Several previous studies investigated differences in executive functions assessed by WCST among patients with schizophrenia, unaffected relatives, and healthy controls. The authors should cite these previous studies and summarize these findings. Furthermore, they mention what is novel in this study.

8. ‘However, in many studies, participants have a greater age at onset.’ Please cite references correctly.

9. ‘Studies have suggested that the earlier the age at onset is, the more severe the executive function impairment.’ Please cite references correctly.

10. ‘Second, consistent with previous research,’ Please cite references correctly.

11. ‘106 subjects were screened by random sampling in our study’ How did the authors select these subjects by random sampling?

12. How many siblings and patients were related?

13. Why did the authors define as clinically stable using HAMD and HAMA but not PANSS? In general, PANSS.

14. ‘The unaffected siblings were the brothers or sisters of the patients. The inclusion

15. criteria for the siblings were as follows: … (3) a history of the long-term use of drugs or mental disorders among the family members.’ Incorrect. The unaffected siblings have a history of schizophrenia among the family members.

16. They should apply ANCOVA with age and sex as covariates but not ANOVA.

17. They should correct for multiple testing.

18. In results ‘The neuropsychological assessment and the study variables were assessed using a one-way analysis of variance (ANOVA) and χ2 tests. When a difference was observed among the three groups, a post hoc comparison using LSD or Tamhane was performed (Table 1).’, ‘To explore whether the WCST performance of the patients and their siblings is related to the neuropsychological tests (MOCA, Stroop test, TMT A and B), we used Pearson’s correlation analyses.’ ‘To explore whether the WCST scores of the patients is related to their Clinical symptoms (PANSS total, PANSS positive, PANSS negative), we used Pearson’s correlation analyses.’ These sentences should be replaced to methods section.

19. ‘Compared with their siblings, we found significant differences in the patients in some variables as follows: SCT [p=0.037] and SWT [p=0.034]; TMT B [p =0.028]), SCT [p= 0.033], SWT [p=0.013], SIT [p=0.001], DSTF (p < 0.001), DSTB (p=0.004), TMT A [p=0.008] and TMT B [p= 0.003] showed differences between the healthy controls and unaffected siblings.’ Please check the sentence carefully. differences in the patients?

20. ‘The data displayed in Table 2 show the results of the WCST.’ Fig2?

21. ‘In any of the three groups, no relationship was found between the WCST performance and the demographic information.(Table 2)’ Table 2?

22. Overall, I did not check these mistakes throughout manuscript because there were too many mistakes.

Reviewer #3: Dear Authors

There is no match between title and content.

The abstract does not fully reflect the content of the work.

In the Abstract there is no decoding of the abbreviated name of the healthy control group

The test procedure lasted 2 hours. How do people with schizophrenia cope with this?

Standard deviations for age are not indicated.

Representing methods is unnecessarily complex.

In the Discussion there are often no references to previous studies. In the Discussion, an inconsistent presentation of thoughts. Jumping from one thought to another.

6. PLOS authors have the option to publish the peer review history of their article (what does this mean?). If published, this will include your full peer review and any attached files.

Reviewer #1: No

Reviewer #2: **Yes: **Kazutaka Ohi

Reviewer #3: No

---

## [Author Response · Author response to Decision Letter 0]

14 Feb 2023

Dear Editors and Reviewers:

Thank you for your letter and for the reviewers ’ comments concerning our manuscript entitled “Study investigating executive function in schizophrenia patients and their unaffected siblings ” (ID: PONE-D-22-29812). Those comments are all valuable and very helpful for revising and improving our paper, as well as the important guiding significance to o ur researches. We have studied comments carefully and have made correction which we hope meet with approval. Revised portion are marked in red in the paper. The main corrections in the paper and the responds to the reviewer’s comments are as flowing:

Responds to the editor ’s comments:

The reviews, which are insightful and interesting, pointed to some unexplained aspects. The authors will notice the reviewers found merits in your study, but also raised several important concerns.

By my own reading, the manuscript still needs a lot of refinement, mostly related to soundness, conciseness and the control of confounding factors. Although this may sound counterintuitive, I am keen to understand authors' claims and keen on reading a refined manuscript. Please note the reviewers' criticism concerning writing style, English, stats and uncontrolled aspects of your study.

Please respond to the comments AND highlight all changes.

Response: Thank you very much for your suggestion. We have revised the manuscript according to the reviewer's comments, We have uploaded the minimum data set as supporting Information files,polished the article, controlled the confounding factors, updated the results, and added relevant documents. The red part represents deletion, and the blue part represents insertion.

Responds to the reviewer’s comments:

Reviewer #1: 

1. Response to comment: (Please enlist a native English speaker or a colleague to edit the manuscript)

Response: Thanks for your suggestion. However, we do invite a friend of us who is a native English speaker from the USA to help polish our article. And we hope the revised manuscript could be acceptable for you.

2. Response to comment: (Although this is a comment for the Abstract, this should be considered thorough the entire text: the authors need to improve readability and conciseness. The Abstract is way too long and both methods and results subsections need to be shortened. This will improve the reading but will also leave the impression of a "punchy" abstract. Please do consider this throughout the text and try to shorten some excessive details (e. g. the first paragraph of Introduction )

Response: We think this is an excellent suggestion. We have explain the change made, including the exact location where the change can be found in the revised manuscript,The results section of the abstract has been reduced(p1,line20-33), The background has also been reduced and adjusted.（p2,line23-25;p3,line 2-5,8-12,16-18;p4,line 1-3,18-20;p5,line 8-10）

3.Response to comment:( In this line, the authors also need to give more details on the fundamentals of their rationale. The Introduction needs to cover (1) definitions, (2) past findings, (3) the main gaps, (4) how the authors are willing to cover these gaps, (5) the main hypotheses and objectives, and (6) a "punchy" statement at the end of the Introduction pointing to the study;)

Response: Thank you for your constructive comments. We made some adjustments to the background, The relevant literature is also supplemented.(p2,line 13-28;p3,line 19-22)

4.Response to comment:( I'd highly suggest the authors to consider a few references for debating the connectivity (e.g., 10.1093/schbul/sbm034 or 10.1016/j.nicl.2014.07.003) and the aspects of physiology (10.1016/j.psychres.2021.114200))

Response: Thank you for your constructive comments.We cite the relevant literature that.

5.Response to comment:(Regarding the Methods, more details are need. For example, the authors need to explain the timeline, screening, confounding variables, the Cronbach values for each scale, all details of the stats. More specifically, details on eligibility is needed: were the patients free of cognitive disorders? (10.3389/fpsyt.2013.00182), had normal visual acuity (10.1016/j.jpsychires.2022.03.014 and 10.1016/j.clinph.2005.06.013), had no previous contact with substances (10.3389/fpsyg.2018.00288), were drug free or what kind of specific meds were taking (10.1016/j.schres.2018.09.002 / 10.31887/DCNS.2006.8.1/bwinklbaur / 10.1016/j.schres.2017.04.016), if BDNF or GDNF levels were collected and why, if they had previous contact with any clinical trial (10.1097/00004714-200012000-00019) and the equivalents for the meds, if any;)

Response:Thank you for your valuable comments, we apologize for our oversight, and we have made corrections in the method section.

6.Response to comment:( Use more illustrative graphs to demonstrate data)

Response: Thank you very much for your valuable comments, We made changes to the chart, as shown in Figure 2.

7.Response to comment:( Use effect sizes and CIs for the analysis;)

Response: Thank you very much for your advice.We have added data about effectsizes in the results section,(p10-12;table1)

8.Response to comment:( Use the corrected p-value)

Response:Thank you for your comments.Standardized effects were calculated using Cohen d method (https://lbecker.uccs.edu/). The signifcance level was set at a 2-tailed P(α=0.05/3) to control for type I errors.

9.Response to comment:( Explain that data are presented as Mean + SD, )

Response:Thank you for your comments, we have made the appropriate corrections, which are shown in the table.

10.Response to comment:( Please bear that boxplots with individuals values are more representative of the data and the sample (10.1038/nmeth.2813)

Response:Thank you for your suggestion, we have produced a boxplots and look forward to your approval.

11.Response to comment:( Details of meds should be place in Methods)

Response:Thank you for your suggestion, we have made adjustments to this section.

12.Response to comment:( Please bear that the use of effect sizes and CIs are needed for the frequentist analyses)

Response:Thanks to your suggestion, we have added effct sizes to the results section.

13.Response to comment:( My main concern regarding the Discussion is the lack of tightening it up to the findings, but most importantly, the lack of references. There are several sentences without a reference. This definitely needs to be rewritten or adjusted;)

Response:Thank you for taking your time and giving us your comments, we have added to the previous studies in the discussion section, and hope to make the quality of the article even better.

14.Response to comment:( Please update the references list using DOI accordingly to the Journal's standards)

Response: Thank you for your positive comments, We have added doi to the references as required by the journal.

Special thanks to you for your good comments. 

Reviewer #2: 

Reviewer #2: 

1.Response to comment:( There were many careless mistakes, such as typos, inappropriate spaces, inappropriate periods and casual terms, throughout manuscript)

Response:Thanks for your advice, We are very sorry for our negligence of these question. We have revised this, and you can see it in the revised draft

2.Response to comment:（The authors should focus on executive functions but not other cognitive functions or cognitive neuropsychological assessments throughout manuscript.）

Response: Thank you very much for your valuable comments. We use WCST to measure executive function,Use digital span test (forward/backward) (DST F/B), Stroop Test (color/word/interference)(SC/W/I/ T),and Trail Making A/B (TMTA/B) to measure relevant functions is represented in the manuscript by neurocognitive tests which can be discussed to some extent in comparison with the WCST results.

3.Response to comment:( The authors should focus on TE among executive functions throughout manuscript as TE was only different among three groups.)

Response: Thank you for your positive comments, The results part of TE is worth focusing on g; we also tried to find the difference between two, because we selected a group with a lower average age, so it is also a point of interest whether the results are more generalizable. Here, we have mentioned in the revised draft

4.Response to comment:( 4. In Abstract ‘the WCST results demonstrate that the total correct (TC), total error (TE), perseverative response (PR) and perseverative error (PE) scores in the SZ group were significantly lower than those in the HC group (TC (p=0.011), TE (p<0.001), PR (p=0.007) and PE (p=0.002)), and compared to the unaffected siblings’, These directions were incorrect, e.g., TE in the SZ group was significantly lower than those in the HC group?? Definitely incorrect.)

Response: Thank you very much for your comments, and sorry for our misrepresentation. We have revised the abstract section based on the comments of all reviewers(p1,line 25-35)

5.Response to comment:( In Abstract ‘Neurological abnormalities lead to abnormal functioning in siblings and patients, suggesting that genetics plays a considerable role in such results.’ The current study did not examine or focus other cognitive function except for executive function.)

Response: Thank you very much for your comments, The main concern of our study was the difference in executive function between patients and unaffected patients, on the basis of which we measured MOCA, other tests were assessed around executive function, and because of the patients' limited energy, we only measured the patients' basic cognitive function on the measurement task

6.Response to comment:( In introduction ‘Most research indicates that executive impairment is a core feature of cognitive impairment [4].’, Please clarify why the author consider executive impairment is a core feature among cognitive impairments.)

Response: Thank you very much for your advice. Cognitive deficits in executive performance, working memory and attention are considered to be core features in patients with schizophrenia, because these deficits are present from the first psychotic episode1. We have reviewed the relevant literature formulation and found that the formulation is not particularly appropriate, so we have adjusted the formulation.

[1] Breton F , A Planté, Legauffre C , et al. The executive control of attention differentiates patients with schizophrenia, their first-degree relatives and healthy controls[J]. Neuropsychologia, 2011, 49(2):203-208.

7.Response to comment:( Several previous studies investigated differences in executive functions assessed by WCST among patients with schizophrenia, unaffected relatives, and healthy controls. The authors should cite these previous studies and summarize these findings. Furthermore, they mention what is novel in this study.)

Response: Special thanks to you for your good comments. In the background section we provide a review of relevant studies on executive functions. And the novelty of our study lies in the fact that we selected a group with a younger mean average age to explore the differences in executive function between patients and unaffected siblings.

8.Response to comment:( However, in many studies, participants have a greater age at onset.’ Please cite references correctly.)

Response: Thank you for your comments, we have made changes to this.

18. Gold JM, Robinson B, Leonard CJ, Hahn B, Chen S, McMahon RP, et al. Selective Attention, Working Memory, and Executive Function as Potential Independent Sources of Cognitive Dysfunction in Schizophrenia. Schizophrenia bulletin. 2018;44: 1227–1234. doi:10.1093/schbul/sbx155

9.Response to comment:(Studies have suggested that the earlier the age at onset is, the more severe the executive function impairment.’ Please cite references correctly.)

Response: Thank you for your comments, We have adjusted the formulation of the paragraph from‘Studies have suggested that the earlier the age at onset is, the more severe the executive function impairment.’ to ‘Studies have suggested that patients with schizophrenia demonstrated similar age-related declines across most neuropsychologic functions ’ .

Sanchez-Torres AM, Basterra V, Moreno-Izco L, Rosa A, Fananas L, Zarzuela A, et al. Executive functioning in schizophrenia spectrum disorder patients and their unaffected siblings: a ten-year follow-up study. Schizophr Res. 2013;143: 291–6. doi:10.1016/j.schres.2012.11.026

Robert Fucetola, Larry J. Seidman, William S. Kremen, Stephen V. Faraone, Jill M. Goldstein, Ming T. Tsuang. Age and Neuropsychologic Function in Schizophrenia: A Decline in Executive Abilities beyond That Observed in Healthy Volunteers. Biol Psychiatry. 2000;48: 137–46. doi:10.1016/s0006-3223(00)00240-7

10.Response to comment:( ‘Second, consistent with previous research,’ Please cite references correctly.)

Response: Thank you for your comments.We have quoted relevant literature.

11.Response to comment:( ‘106 subjects were screened by random sampling in our study’ How did the authors select these subjects by random sampling? ’)

Response: Thank you for your comments, Inclusion or exclusion of groups by random assignment using the random number method among outpatients.

12.Response to comment:(How many siblings and patients were related?)

Response: Thank you for your comments, in our study, A total of 25 family members have relatives with patients.

13.Response to comment:( Why did the authors define as clinically stable using HAMD and HAMA but not PANSS? In general, PANSS.)

Response: Thank you for your valuable comments, in this study, we are collecting patients in psychiatric outpatient clinic, and most of the patients in the outpatient clinic are in the stable stage of the disease, so we will have a professional physician to assess the psychiatric symptoms using PANSS before the start of the experiment, and also to ensure the accuracy of the assessment results, we must ensure that the emotional state of the patients and other participants is stable, so we have chosen HAMA,HAMD in the study.

14.Response to comment:( ‘The unaffected siblings were the brothers or sisters of the patients. The inclusion)

Response: Thank you for your comments, the unaffected sibling in our study was the patient's brother or sister.

15.Response to comment:( criteria for the siblings were as follows: … (3) a history of the long-term use of drugs or mental disorders among the family members.’ Incorrect. The unaffected siblings have a history of schizophrenia among the family members.)

Response: Thank you for your valuable comments, there is a problem with our presentation. ‘a history of the long-term use of drugs or mental disorders among the family members.’ Modify to ‘a history of the long-term use of drugs or mental disorders.’ Because family members with mental illness are not able to be included in the experiment.

16.Response to comment:( They should apply ANCOVA with age and sex as covariates but not ANOVA.)

Response: Thanks for your reminder, we have corrected the statistical method and updated the relevant data(p10,line 4-9).

17.Response to comment:( They should correct for multiple testing)

Response: Thank you for your comments, we have made adjustments to the statistical methods section to ensure the accuracy of the results.(p10,line 1-15)

18.Response to comment:( In results ‘The neuropsychological assessment and the study variables were assessed using a one-way analysis of variance (ANOVA) and χ2 tests. When a difference was observed among the three groups, a post hoc comparison using LSD or Tamhane was performed (Table 1).’, ‘To explore whether the WCST performance of the patients and their siblings is related to the neuropsychological tests (MOCA, Stroop test, TMT A and B), we used Pearson’s correlation analyses.’ ‘To explore whether the WCST scores of the patients is related to their Clinical symptoms (PANSS total, PANSS positive, PANSS negative), we used Pearson’s correlation analyses.’ These sentences should be replaced to methods section.)

Response: Thank you for your suggestions, we have made adjustments to these sections.

19.Response to comment:( ‘Compared with their siblings, we found significant differences in the patients in some variables as follows: SCT [p=0.037] and SWT [p=0.034]; TMT B [p =0.028]), SCT [p= 0.033], SWT [p=0.013], SIT [p=0.001], DSTF (p < 0.001), DSTB (p=0.004), TMT A [p=0.008] and TMT B [p= 0.003] showed differences between the healthy controls and unaffected siblings.’ Please check the sentence carefully. differences in the patients?)

Response: Thank you for your suggestion, we have corrected this.(p11,line 8-13)

20.Response to comment:( ‘The data displayed in Table 2 show the results of the WCST.’ Fig2?)

Response: We are very sorry for the trouble caused to your review, Fig2 may have a problem when uploading the file, we will pay attention to this problem in the revised manuscript, Fig2 shows the difference between the three groups of subjects.

21.Response to comment:( ‘In any of the three groups, no relationship was found between the WCST performance and the demographic information.(Table 2)’ Table 2?)

Response: Thank you for your suggestion, the demographic data analysis is shown in table3, the sentence was misrepresented and we have removed it.

22.Response to comment:( Overall, I did not check these mistakes throughout manuscript because there were too many mistakes.)

Response: Thank you for your comments on the manuscript during your busy schedule, and we apologize for our mistakes. At the same time, we are doing our best to improve the manuscript.

Reviewer #3:

Thank you for your precious opinions

1. Response to comment: (There is no match between title and content.)

Response:Thanks for your kind suggest. The content of the manuscript is the performance of the subjects on WSCT, as well as the results on stroop and number span test, which belong to the category of executive function, so the topic shows executive function.Thank you very much for your comments, we have discussed accordingly, but the manuscript title may still use the original title, in the future we will try to modify the title if necessary, and we have also made adjustments to the content part .

2. Response to comment: (The abstract does not fully reflect the content of the work. )

Response: Thank you very much for your comments. We have made adjustments to the abstract based on your comments and those of other reviewers, and we sincerely thank you for pointing out our mistakes.

3. Response to comment: (In the Abstract there is no decoding of the abbreviated name of the healthy control group )

Response: Thanks to your valuable suggestion, we use HCs in the abstract to denote healthy controls.

4. Response to comment: (The test procedure lasted 2 hours. How do people with schizophrenia cope with this? )

Response: Thanks for your suggest. In the process of completing wsct tasks, it should be completed at one time; However, other tests will give patients enough time to have a short rest in between, in order to show a more realistic level.

5. Response to comment: (Standard deviations for age are not indicated.)

Response: Thanks for your suggest. See table1 for the standard deviation of age.

6. Response to comment: (Representing methods is unnecessarily complex. )

Response:Thanks to your suggestion, vwe have scaled down the methods section.

7. Response to comment: (In the Discussion there are often no references to previous studies. In the Discussion, an inconsistent presentation of thoughts. Jumping from one thought to another. )

Response: Thank you for taking your time and giving us your comments, we have added to the previous studies in the discussion section, and hope to make the quality of the article even better

---

## [Decision Letter · Decision Letter 1]

13 Mar 2023

PONE-D-22-29812R1Study investigating executive function in schizophrenia patients and their unaffected siblingsPLOS ONE

Dear Dr. xian,

Thank you for submitting your manuscript to PLOS ONE. After careful consideration, we feel that it has merit but does not fully meet PLOS ONE’s publication criteria as it currently stands. Therefore, we invite you to submit a revised version of the manuscript that addresses the points raised during the review process.

I call out authors’ attention to all reviewers’ comment. But, important as well, to Rev #2 who raised several concerns about the rigour of your Methods (the lack of rigour), eligibility criteria and stats. Please carefully look for studies on the matter that observed illness duration and medication use - since this is *essential to control”. You will find lots of studies on perceptual and high-level that looked this - check their Methods and criteria, not the theme per se.

We look forward to receiving your revised manuscript.

Kind regards,

Thiago P. Fernandes, PhD

Academic Editor

PLOS ONE

Reviewers' comments:

Reviewer's Responses to Questions

**Comments to the Author**

1. If the authors have adequately addressed your comments raised in a previous round of review and you feel that this manuscript is now acceptable for publication, you may indicate that here to bypass the “Comments to the Author” section, enter your conflict of interest statement in the “Confidential to Editor” section, and submit your "Accept" recommendation.

Reviewer #1: (No Response)

Reviewer #2: (No Response)

Reviewer #3: All comments have been addressed

2. Is the manuscript technically sound, and do the data support the conclusions?

Reviewer #1: Yes

Reviewer #2: No

Reviewer #3: Yes

3. Has the statistical analysis been performed appropriately and rigorously? 

Reviewer #1: Yes

Reviewer #2: I Don't Know

Reviewer #3: Yes

4. Have the authors made all data underlying the findings in their manuscript fully available?

Reviewer #1: Yes

Reviewer #2: No

Reviewer #3: Yes

5. Is the manuscript presented in an intelligible fashion and written in standard English?

Reviewer #1: Yes

Reviewer #2: No

Reviewer #3: Yes

6. Review Comments to the Author

Reviewer #1: Thank you for addressing reviewers’ concerns. The manuscript is well-written and much better. But I haven’t found the suggested references from my previous reading - to be on Methods and eligibility criteria. Besides, consider Hedges’ g instead of Cohen’s because of the sample size and deviations from kurtosis and skewness. Please correct.

Reviewer #2: (No Response)

Reviewer #3: The authors have revised and improved the article. They have studied comments carefully and have made

correction. They have uploaded the minimum data set as supporting Information files, controlled the confounding factors, updated the results, and added relevant documents, made corrections in the method section, results and discussion.

7. PLOS authors have the option to publish the peer review history of their article (what does this mean?). If published, this will include your full peer review and any attached files.

Reviewer #1: No

Reviewer #2: No

Reviewer #3: **Yes: **Irina Shoshina

---

## [Author Response · Author response to Decision Letter 1]

6 Apr 2023

Dear Editors and Reviewers:

Thank you for your letter and for the reviewers ’ comments concerning our manuscript entitled “Study investigating executive function in schizophrenia patients and their unaffected siblings ” (ID: PONE-D-22-29812R2). Those comments are all valuable and very helpful for revising and improving our paper, as well as the important guiding significance to our researches. We have studied comments carefully and have made correction which we hope meet with approval. Revised portion are marked in red in the paper. The main corrections in the paper and the responds to the reviewer’s comments are as flowing:

Responds to the editor ’s comments:

Thank you for submitting your manuscript to PLOS ONE. After careful consideration, we feel that it has merit but does not fully meet PLOS ONE’s publication criteria as it currently stands. Therefore, we invite you to submit a revised version of the manuscript that addresses the points raised during the review process.

I call out authors’ attention to all reviewers’ comment. But, important as well, to Rev #2 who raised several concerns about the rigour of your Methods (the lack of rigour), eligibility criteria and stats. Please carefully look for studies on the matter that observed illness duration and medication use - since this is *essential to control”. You will find lots of studies on perceptual and high-level that looked this - check their Methods and criteria, not the theme per se.

Response: Thank you very much for your suggestion. We have revised the manuscript according to the reviewer's comments, We have uploaded the minimum data set as supporting Information files, We have controlled for confounding factors and used more rigorous statistical methods to ensure the accuracy of the results. We have supplemented the duration of the disease and drug use in the Methods section and Table 1, The red part represents the modified content.

Responds to the reviewer’s comments:

Reviewer #1: 

Thank you for addressing reviewers’ concerns. The manuscript is well-written and much better. But I haven’t found the suggested references from my previous reading - to be on Methods and eligibility criteria. Besides, consider Hedges’ g instead of Cohen’s because of the sample size and deviations from kurtosis and skewness. Please correct.

1. Response to comment: 

Thank you for your comments.

Based on these observations and in combination with our research, we have cited relevant literature 10.3389/fpsyt.2013.00182,10.1016/j.jpsychires.2022.03.014,10.3389/fpsyg.2018.00288, 10.1097/00004714-200012000-00019.besides, we use Hedges' g instead of Cohen's value.

Reviewer #2: 

We will respond based on your comments last time, as there are no new comments for this modification. We uploaded research data and made modifications to the methodology, citing previous research reports, controlling for confounding factors, and making adjustments to drug use.(p4-7)

10. Y M, K O. Cortical gyrification in schizophrenia: current perspectives. Neuropsychiatric disease and treatment. 2018;14. doi:10.2147/NDT.S145273

20. Ohi K, Nemoto K, Kataoka Y, Sugiyama S, Muto Y, Shioiri T, et al. Alterations in hippocampal subfield volumes among schizophrenia patients, their first-degree relatives and healthy subjects. Prog Neuropsychopharmacol Biol Psychiatry. 2021;110: 110291. doi:10.1016/j.pnpbp.2021.110291

1.Response to comment:( There were many careless mistakes, such as typos, inappropriate spaces, inappropriate periods and casual terms, throughout manuscript)

Response:Thanks for your advice, We are very sorry for our negligence of these question. We have revised this, and you can see it in the revised draft

2.Response to comment:（The authors should focus on executive functions but not other cognitive functions or cognitive neuropsychological assessments throughout manuscript.）

Response: Thank you very much for your valuable comments. We use WCST to measure executive function,Use digital span test (forward/backward) (DST F/B), Stroop Test (color/word/interference)(SC/W/I/ T),and Trail Making A/B (TMTA/B) to measure relevant functions is represented in the manuscript by neurocognitive tests which can be discussed to some extent in comparison with the WCST results.

3.Response to comment:( The authors should focus on TE among executive functions throughout manuscript as TE was only different among three groups.)

Response: Thank you for your positive comments, The results part of TE is worth focusing on g; we also tried to find the difference between two, because we selected a group with a lower average age, so it is also a point of interest whether the results are more generalizable. Here, we have mentioned in the revised draft

4.Response to comment:( 4. In Abstract ‘the WCST results demonstrate that the total correct (TC), total error (TE), perseverative response (PR) and perseverative error (PE) scores in the SZ group were significantly lower than those in the HC group (TC (p=0.011), TE (p<0.001), PR (p=0.007) and PE (p=0.002)), and compared to the unaffected siblings’, These directions were incorrect, e.g., TE in the SZ group was significantly lower than those in the HC group?? Definitely incorrect.)

Response: Thank you very much for your comments, and sorry for our misrepresentation. We have revised the abstract section based on the comments of all reviewers(p1,line 25-35)

5.Response to comment:( In Abstract ‘Neurological abnormalities lead to abnormal functioning in siblings and patients, suggesting that genetics plays a considerable role in such results.’ The current study did not examine or focus other cognitive function except for executive function.)

Response: Thank you very much for your comments, The main concern of our study was the difference in executive function between patients and unaffected patients, on the basis of which we measured MOCA, other tests were assessed around executive function, and because of the patients' limited energy, we only measured the patients' basic cognitive function on the measurement task

6.Response to comment:( In introduction ‘Most research indicates that executive impairment is a core feature of cognitive impairment [4].’, Please clarify why the author consider executive impairment is a core feature among cognitive impairments.)

Response: Thank you very much for your advice. Cognitive deficits in executive performance, working memory and attention are considered to be core features in patients with schizophrenia, because these deficits are present from the first psychotic episode1. We have reviewed the relevant literature formulation and found that the formulation is not particularly appropriate, so we have adjusted the formulation.

[1] Breton F , A Planté, Legauffre C , et al. The executive control of attention differentiates patients with schizophrenia, their first-degree relatives and healthy controls[J]. Neuropsychologia, 2011, 49(2):203-208.

7.Response to comment:( Several previous studies investigated differences in executive functions assessed by WCST among patients with schizophrenia, unaffected relatives, and healthy controls. The authors should cite these previous studies and summarize these findings. Furthermore, they mention what is novel in this study.)

Response: Special thanks to you for your good comments. In the background section we provide a review of relevant studies on executive functions. And the novelty of our study lies in the fact that we selected a group with a younger mean average age to explore the differences in executive function between patients and unaffected siblings.

8.Response to comment:( However, in many studies, participants have a greater age at onset.’ Please cite references correctly.)

Response: Thank you for your comments, we have made changes to this.

18. Gold JM, Robinson B, Leonard CJ, Hahn B, Chen S, McMahon RP, et al. Selective Attention, Working Memory, and Executive Function as Potential Independent Sources of Cognitive Dysfunction in Schizophrenia. Schizophrenia bulletin. 2018;44: 1227–1234. doi:10.1093/schbul/sbx155

9.Response to comment:(Studies have suggested that the earlier the age at onset is, the more severe the executive function impairment.’ Please cite references correctly.)

Response: Thank you for your comments, We have adjusted the formulation of the paragraph from‘Studies have suggested that the earlier the age at onset is, the more severe the executive function impairment.’ to ‘Studies have suggested that patients with schizophrenia demonstrated similar age-related declines across most neuropsychologic functions ’ .

Sanchez-Torres AM, Basterra V, Moreno-Izco L, Rosa A, Fananas L, Zarzuela A, et al. Executive functioning in schizophrenia spectrum disorder patients and their unaffected siblings: a ten-year follow-up study. Schizophr Res. 2013;143: 291–6. doi:10.1016/j.schres.2012.11.026

Robert Fucetola, Larry J. Seidman, William S. Kremen, Stephen V. Faraone, Jill M. Goldstein, Ming T. Tsuang. Age and Neuropsychologic Function in Schizophrenia: A Decline in Executive Abilities beyond That Observed in Healthy Volunteers. Biol Psychiatry. 2000;48: 137–46. doi:10.1016/s0006-3223(00)00240-7

10.Response to comment:( ‘Second, consistent with previous research,’ Please cite references correctly.)

Response: Thank you for your comments.We have quoted relevant literature.

11.Response to comment:( ‘106 subjects were screened by random sampling in our study’ How did the authors select these subjects by random sampling? ’)

Response: Thank you for your comments, Inclusion or exclusion of groups by random assignment using the random number method among outpatients.

12.Response to comment:(How many siblings and patients were related?)

Response: Thank you for your comments, in our study, A total of 25 family members have relatives with patients.

13.Response to comment:( Why did the authors define as clinically stable using HAMD and HAMA but not PANSS? In general, PANSS.)

Response: Thank you for your valuable comments, in this study, we are collecting patients in psychiatric outpatient clinic, and most of the patients in the outpatient clinic are in the stable stage of the disease, so we will have a professional physician to assess the psychiatric symptoms using PANSS before the start of the experiment, and also to ensure the accuracy of the assessment results, we must ensure that the emotional state of the patients and other participants is stable, so we have chosen HAMA,HAMD in the study.

14.Response to comment:( ‘The unaffected siblings were the brothers or sisters of the patients. The inclusion)

Response: Thank you for your comments, the unaffected sibling in our study was the patient's brother or sister.

15.Response to comment:( criteria for the siblings were as follows: … (3) a history of the long-term use of drugs or mental disorders among the family members.’ Incorrect. The unaffected siblings have a history of schizophrenia among the family members.)

Response: Thank you for your valuable comments, there is a problem with our presentation. ‘a history of the long-term use of drugs or mental disorders among the family members.’ Modify to ‘a history of the long-term use of drugs or mental disorders.’ Because family members with mental illness are not able to be included in the experiment.

16.Response to comment:( They should apply ANCOVA with age and sex as covariates but not ANOVA.)

Response: Thanks for your reminder, we have corrected the statistical method and updated the relevant data(p10,line 4-9).

17.Response to comment:( They should correct for multiple testing)

Response: Thank you for your comments, we have made adjustments to the statistical methods section to ensure the accuracy of the results.(p10,line 1-15)

18.Response to comment:( In results ‘The neuropsychological assessment and the study variables were assessed using a one-way analysis of variance (ANOVA) and χ2 tests. When a difference was observed among the three groups, a post hoc comparison using LSD or Tamhane was performed (Table 1).’, ‘To explore whether the WCST performance of the patients and their siblings is related to the neuropsychological tests (MOCA, Stroop test, TMT A and B), we used Pearson’s correlation analyses.’ ‘To explore whether the WCST scores of the patients is related to their Clinical symptoms (PANSS total, PANSS positive, PANSS negative), we used Pearson’s correlation analyses.’ These sentences should be replaced to methods section.)

Response: Thank you for your suggestions, we have made adjustments to these sections.

19.Response to comment:( ‘Compared with their siblings, we found significant differences in the patients in some variables as follows: SCT [p=0.037] and SWT [p=0.034]; TMT B [p =0.028]), SCT [p= 0.033], SWT [p=0.013], SIT [p=0.001], DSTF (p < 0.001), DSTB (p=0.004), TMT A [p=0.008] and TMT B [p= 0.003] showed differences between the healthy controls and unaffected siblings.’ Please check the sentence carefully. differences in the patients?)

Response: Thank you for your suggestion, we have corrected this.(p11,line 8-13)

20.Response to comment:( ‘The data displayed in Table 2 show the results of the WCST.’ Fig2?)

Response: We are very sorry for the trouble caused to your review, Fig2 may have a problem when uploading the file, we will pay attention to this problem in the revised manuscript, Fig2 shows the difference between the three groups of subjects.

21.Response to comment:( ‘In any of the three groups, no relationship was found between the WCST performance and the demographic information.(Table 2)’ Table 2?)

Response: Thank you for your suggestion, the demographic data analysis is shown in table3, the sentence was misrepresented and we have removed it.

22.Response to comment:( Overall, I did not check these mistakes throughout manuscript because there were too many mistakes.)

Response: Thank you for your comments on the manuscript during your busy schedule, and we apologize for our mistakes. At the same time, we are doing our best to improve the manuscript.

Thank you very much for your comments. The quality of our manuscript has been improved, and we hope to receive your recognition

Reviewer #3:

Thank you for your precious opinions

 Response to comment: (The authors have revised and improved the article. They have studied comments carefully and have madecorrection. They have uploaded the minimum data set as supporting Information files, controlled the confounding factors, updated the results, and added relevant documents, made corrections in the method section, results and discussion.)

Response: Thank you very much for your comments. Due to your suggestions, the quality of our manuscript has been improved. On this basis, we have added more valuable research literature. Thank you very much for your valuable comments.

---

## [Editor Report · Decision Letter 2]

14 Apr 2023

Study investigating executive function in schizophrenia patients and their unaffected siblings

PONE-D-22-29812R2

Dear Dr. xian,

We’re pleased to inform you that your manuscript has been judged scientifically suitable for publication and will be formally accepted for publication once it meets all outstanding technical requirements.

Kind regards,

Thiago P. Fernandes, PhD

Academic Editor

PLOS ONE

Additional Editor Comments (optional):

Why the p-values are followed by superscript numbers 10^supercript? Please correct this.

Also, place the CIs for the Hedges' g in text and remember do capitalise the 'h' for 'Hedges". The effect sizes should be followed by the stats. Example: [stat = value, p = value, Hedges' = value (CIs)].

Remember to organise the sections and a few wording.

This would be easily fixed during typesetting.

Wishing you success with your study.

Thank you for your submission and valuable edits.
---

## [Editor Report · Acceptance letter]

18 Apr 2023

PONE-D-22-29812R2 

Study investigating executive function in schizophrenia patients And their unaffected siblings 

Dear Dr. xian:

I'm pleased to inform you that your manuscript has been deemed suitable for publication in PLOS ONE. Congratulations! Your manuscript is now with our production department. 

Kind regards, 

on behalf of

Dr. Thiago P. Fernandes 

Academic Editor

PLOS ONE